

# VGEA: an RNA viral assembly toolkit

Paul E. Oluniyi[1,2], Fehintola Ajogbasile[1,2], Judith Oguzie[1,2], Jessica Uwanibe[1,2], Adeyemi Kayode[1,2], Anise Happi[2], Alphonsus Ugwu[1,2], Testimony Olumade[1,2], Olusola Ogunsanya[3], Philomena Ehiaghe Eromon[2], Onikepe Folarin[1,2], Simon D.W. Frost[4,5], Jonathan Heeney[6] and Christian T. Happi[1,2]

[1] Department of Biological Sciences, Faculty of Natural Sciences, Redeemer's University, Ede, Osun, Nigeria
[2] African Centre of Excellence for Genomics of Infectious Diseases (ACEGID), Redeemer's University, Ede, Osun, Nigeria
[3] Department of Veterinary Pathology, Faculty of Veterinary Medicine, University of Ibadan, Ibadan, Oyo, Nigeria
[4] Microsoft Research, Redmond, WA, United States of America
[5] London School of Hygiene & Tropical Medicine, London, United Kingdom
[6] Department of Veterinary Medicine, University of Cambridge, Cambridge, United Kingdom

Corresponding author
Christian T. Happi,
happic@run.edu.ng

## ABSTRACT

Next generation sequencing (NGS)-based studies have vastly increased our understanding of viral diversity. Viral sequence data obtained from NGS experiments are a rich source of information, these data can be used to study their epidemiology, evolution, transmission patterns, and can also inform drug and vaccine design. Viral genomes, however, represent a great challenge to bioinformatics due to their high mutation rate and forming quasispecies in the same infected host, bringing about the need to implement advanced bioinformatics tools to assemble consensus genomes well-representative of the viral population circulating in individual patients. Many tools have been developed to preprocess sequencing reads, carry-out *de novo* or reference-assisted assembly of viral genomes and assess the quality of the genomes obtained. Most of these tools however exist as standalone workflows and usually require huge computational resources. Here we present (**V**iral **G**enomes **E**asily **A**nalyzed), a Snakemake workflow for analyzing RNA viral genomes. VGEA enables users to map sequencing reads to the human genome to remove human contaminants, split bam files into forward and reverse reads, carry out *de novo* assembly of forward and reverse reads to generate contigs, pre-process reads for quality and contamination, map reads to a reference tailored to the sample using corrected contigs supplemented by the user's choice of reference sequences and evaluate/compare genome assemblies. We designed a project with the aim of creating a flexible, easy-to-use and all-in-one pipeline from existing/stand-alone bioinformatics tools for viral genome analysis that can be deployed on a personal computer. VGEA was built on the Snakemake workflow management system and utilizes existing tools for each step: **fastp** (*Chen et al., 2018*) for read trimming and read-level quality control, **BWA** (*Li & Durbin, 2009*) for mapping sequencing reads to the human reference genome, **SAMtools** (*Li et al., 2009*) for extracting unmapped reads and also for splitting bam files into fastq files, **IVA** (*Hunt et al., 2015*) for *de novo* assembly to generate contigs, **shiver** (*Wymant et al., 2018*) to pre-process reads for quality and contamination, then map to a reference tailored to the sample using corrected contigs supplemented with the user's choice of existing reference sequences, **SeqKit** (*Shen et al., 2016*) for cleaning shiver assembly for QUAST, **QUAST** (*Gurevich et al., 2013*) to evaluate/assess the quality of genome assemblies and **MultiQC**

(*Ewels et al., 2016*) for aggregation of the results from fastp, BWA and QUAST. Our pipeline was successfully tested and validated with SARS-CoV-2 ($n = 20$), HIV-1 ($n = 20$) and Lassa Virus ($n = 20$) datasets all of which have been made publicly available. VGEA is freely available on GitHub at: https://github.com/pauloluniyi/VGEA under the GNU General Public License.

## INTRODUCTION

The most abundant biological entities on Earth are viruses as they can be found among all cellular forms of life. So far, over four thousand five hundred viral species have been discovered, from which a huge amount of sequence information has been collected by researchers and scientists all over the world (*Pickett et al., 2012*; *Sharma, Priyadarshini & Vrati, 2015*; *Brister et al., 2015*). In recent times (past two decades), a number of these viruses have emerged in the human population causing disease outbreaks and sometimes pandemics. These viruses include mainly: Influenza virus, Severe Acute Respiratory Syndrome (SARS) coronavirus, Middle East Respiratory Syndrome (MERS) coronavirus, Ebola virus, Yellow fever virus, Lassa virus (LASV), Zika virus (*Chan, 2002*; *Bean et al., 2013*; *Folarin et al., 2016*; *Grubaugh et al., 2017*; *Metsky et al., 2017*; *Siddle et al., 2018*; *Ajogbasile et al., 2020*) and SARS-CoV-2 (*Chen et al., 2020*; *Holshue et al., 2020*; *Sohrabi et al., 2020*). During these outbreaks and pandemics, genomic sequencing for identification and characterization of the transmission and evolution of the causative agents have proved to be critical in helping inform disease surveillance and epidemiology.

Next Generation Sequencing (NGS) platforms have been widely accepted as high-throughput, open view technologies that have many attractive features for virus detection and assembly (*Tang & Chiu, 2010*; *Mokili, Rohwer & Dutilh, 2012*). NGS-based studies have vastly increased our understanding of viral diversity (*Reyes et al., 2010*; *Cantalupo et al., 2011*). Pathogen sequence data obtained from NGS experiments are a rich source of information, these data can be used to study their epidemiology, evolution, transmission patterns, and can also inform drug and vaccine design. The field of genomics, especially pathogen genomics has been transformed by NGS, with costs constantly decreasing, equipment becoming more portable/field deployable during outbreaks and remarkable increase in data availability.

The huge amount of data being generated requires various processing steps such as removal of primers and adapters, quality filtering and control which is usually crucial for various downstream analysis. Several tools have been developed for these purposes, such as fastp (*Chen et al., 2018*) and Trimmomatic (*Bolger, Lohse & Usadel, 2014*).

Reconstructing viral genomes from NGS data is usually achieved through *de novo* assembly (which is the process of assembling genomes using overlapping sequencing reads), or through a reference-guided approach (which involves mapping sequence reads to a reference genome). Numerous tools have been developed for these purposes; SPAdes

(*Bankevich et al., 2012*), Burrows-Wheeler Alignment tool (BWA), V-GAP (*Nakamura et al., 2016*), VirusTAP (*Yamashita, Sekizuka & Kuroda, 2016*), V-Pipe (*Posada-Céspedes et al., 2021*) and viral-ngs (https://github.com/broadinstitute/viral-ngs), amongst others. Contigs generated by *de novo* assembly however do not provide a complete summary of reads, misassembly can result in the contigs having an incorrect structure, and for parts of the genome where contigs could not be assembled, no information is available. In addition, reference-guided assembly of viral genomes can lead to biased loss of information which can then skew epidemiological and evolutionary conclusions (*Wymant et al., 2018*).

Variant analysis and genome quality assessment to detect variants and changes occurring across the genome of a virus is also a key step in viral genome analysis as viruses (especially RNA viruses) are known to have high mutation rates (*Duffy, 2018*). Variant analysis is important for detecting outbreak origins and for phylogenetic/phylogeographic studies and best practices for variant identification in microbial genomes have been proposed in literature and adopted to a large extent (*Van der Auwera et al., 2013*).

A number of pipelines that have been developed for downstream analysis of viral genomes require high performance computing (HPC) clusters and/or cloud-based systems *e.g.*, the V-pipe authors recommend running V-pipe on clusters because for most applications, running V-pipe on a local machine may not be efficient (https://github.com/cbg-ethz/V-pipe/wiki/advanced) and some of these pipelines are only web-based such as VirAmp (*Wan et al., 2015*) and VirusTAP (*Yamashita, Sekizuka & Kuroda, 2016*). Also, some pipelines have many dependencies to be installed especially if the analysis requires multiple tasks to be performed. In low-and-middle income countries (LMICs) where most scientists do not have access to HPC clusters or cloud-based systems and where internet connection is too unstable to regularly make use of web-based platforms for analysis, this can be a daunting task.

The challenges listed above motivated the development of VGEA (Viral Genomes Easily Analyzed, available online at https://github.com/pauloluniyi/VGEA). VGEA makes use of existing bioinformatics pipeline/tools to carry out various viral genome analysis tasks and is built on an advanced workflow management system, Snakemake (*Köster & Rahmann, 2012*).

## MATERIALS AND METHODS

### Datasets

We successfully tested and validated VGEA with SARS-CoV-2 ($n = 20$) and Lassa Virus ($n = 20$) datasets sequenced on the illumina MiSeq and illumina FGx sequencing machines in our laboratory at the African Centre of Excellence for Genomics of Infectious Diseases (ACEGID), Redeemer's University, Ede, Nigeria. Briefly, samples were inactivated in buffer AVL and viral RNA was extracted according to the QiAmp viral RNA mini kit (Qiagen) manufacturer's instructions. Extracted RNA was treated with Turbo DNase to remove contaminating DNA, followed by cDNA synthesis with random hexamers. Sequencing libraries were prepared using the Nextera XT kit (Illumina) as previously described (*Matranga et al., 2016*) and sequenced on the Illumina Miseq platform with

101 base pair paired-end reads. We also tested and validated VGEA with HIV-1 datasets sequenced on the illumina HiSeq 2500 obtained from NCBI Sequence Read Archive (SRA). We made use of 60 test datasets (Lassa Virus (20), SARS-CoV-2 (20) and HIV-1 (20)) for the validation of the VGEA pipeline. All our test datasets are available on figshare (https://doi.org/10.6084/m9.figshare.13009997).

**Implementation**

The installation of VGEA requires the pipeline to be downloaded onto a personal computer and creation of a conda environment to set up all dependencies. Complete installation steps are in the github README file: https://github.com/pauloluniyi/VGEA/blob/master/README.md

The analysis of VGEA is broken down into a set of 'rules' that links the output file of an analysis into the input of the next task in the general workflow (Fig. 1). The dependencies are **fastp** for read trimming and read-level quality control, **BWA** for mapping sequencing reads to the human reference genome, **SAMtools** for extracting unmapped reads and also for splitting bam files into fastq files, **IVA** for *de novo* assembly to generate contigs, **shiver** to pre-process reads for quality and contamination, then map to a reference tailored to the sample using corrected contigs supplemented with the user's choice of existing reference sequences, **SeqKit** for cleaning shiver assembly for QUAST, **QUAST** to evaluate/assess the quality of genome assemblies and **MultiQC** for aggregation of the results from fastp, BWA and QUAST

All of these tools can be installed using a bioconda channel (*Grüning et al., 2018*). The input files for VGEA are paired-end fastq files. VGEA allows full customization of the pipeline, so users can modify the parameters used in running their samples. It is possible to modify every step of the workflow to suit the samples being processed. Users can also add more steps to the pipeline as they see fit. The pipeline runs on Linux/Unix and Mac. However, no prior programming is required to run the pipeline and, once the user supplies the input, the whole workflow can run automatically from beginning to end.

# RESULTS

VGEA carries out read trimming and quality control tasks on input FASTQ data using fastp (Fig. 2). This increases the quality of data used for subsequent steps of the pipeline. VGEA then maps reads to the human reference genome in order to remove human contaminants, the pipeline carries out this step using BWA. Genome assembly and consensus sequence generation is carried out, together with the generation of summary minority-variant information (base frequencies at each position) and detailed minority-variant information (all reads aligned to their correct position in the genome). VGEA carries out assembly using **IVA** and generates consensus sequences using **shiver**. Previous study by the shiver developers has shown the systematic superiority of mapping to shiver's constructed reference compared with mapping the same reads to the closest of 3,249 references: median values of 13 bases called differently and more accurately, zero bases called differently and less accurately, and 205 bases of missing sequence recovered (*Wymant et al., 2018*).
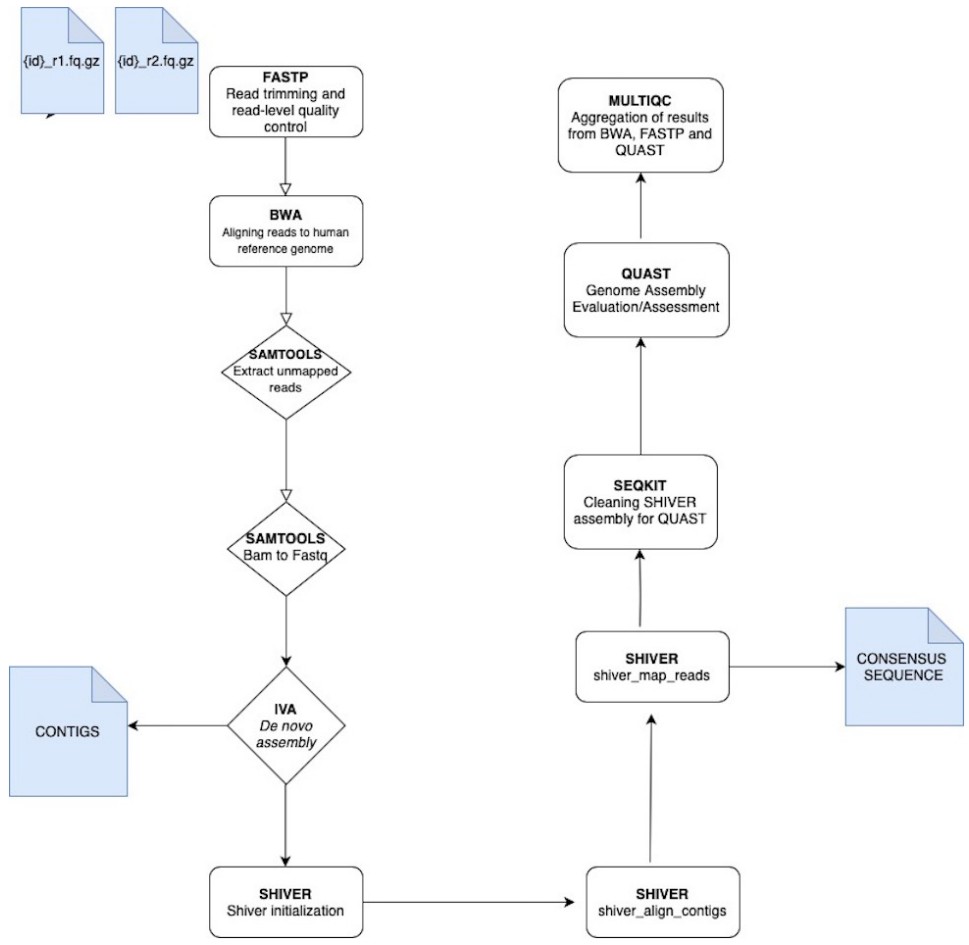

**Figure 1 A schematic workflow of VGEA.** User-supplied paired-end fastq files are pre-processed and trimmed using **FASTP** followed by mapping to the human reference genome with **BWA**. Following mapping, a BAM file containing unaligned/unmapped reads is extracted using **SAMTOOLS**. This BAM file is then split into fastq files of forward and reverse reads also with **SAMTOOLS** after which *de novo* assembly is carried out using **IVA**. Following *de novo* assembly, **SHIVER** is used to map the reads and generate consensus sequences, and detailed minority variant information (full explanation of the shiver method is in File S1). **SEQKIT** is used to clean the SHIVER output for QUAST after which genome evaluation and assessment is carried out using **QUAST**. **MULTIQC** is then used for aggregation of results from BWA, FASTP and QUAST.

VGEA also assesses the quality of genome assemblies using QUAST. QUAST evaluates metrics such as contig sizes, misassemblies and structural variations, genome representation and its functional elements, variations of N50 based on aligned blocks and then presents these statistics in graphical form. QUAST also makes a histogram of several metrics including the number of complete genes, operons and the genome fraction (%). Finally, VGEA compiles the results of BWA, fastp and QUAST into a single MultiQC report (Fig. 3).

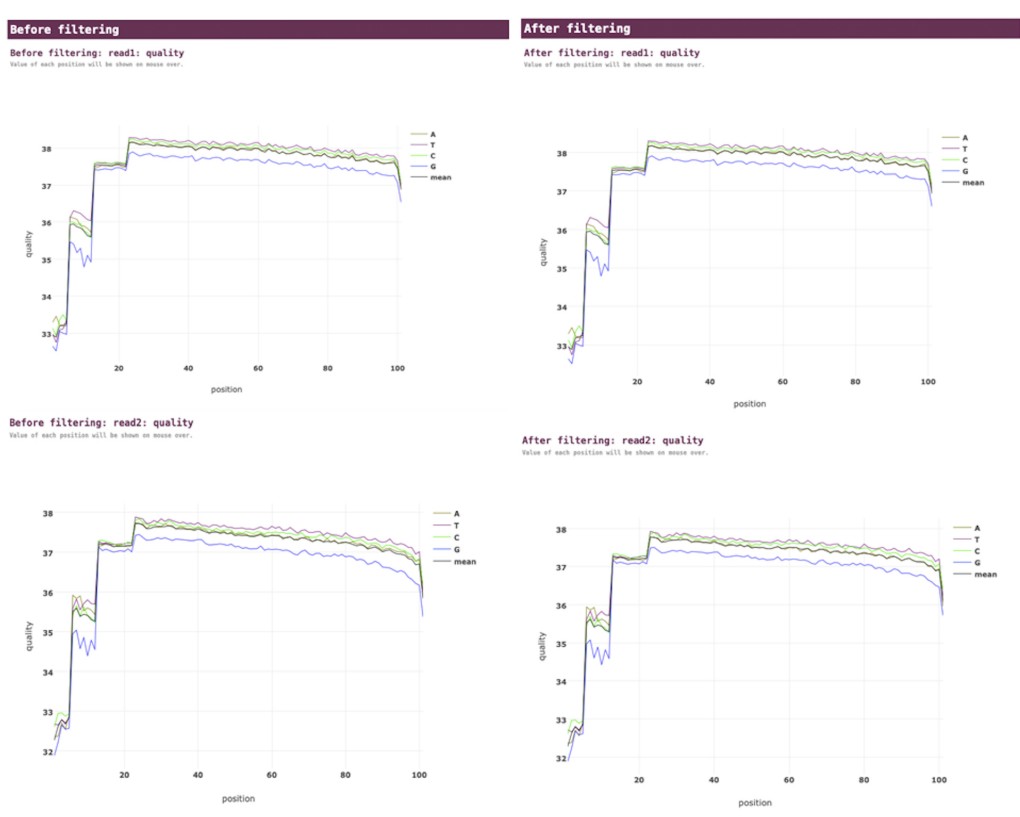

**Figure 2** Fastp pre-processing report for a SARS-CoV-2 test dataset analyzed using VGEA.

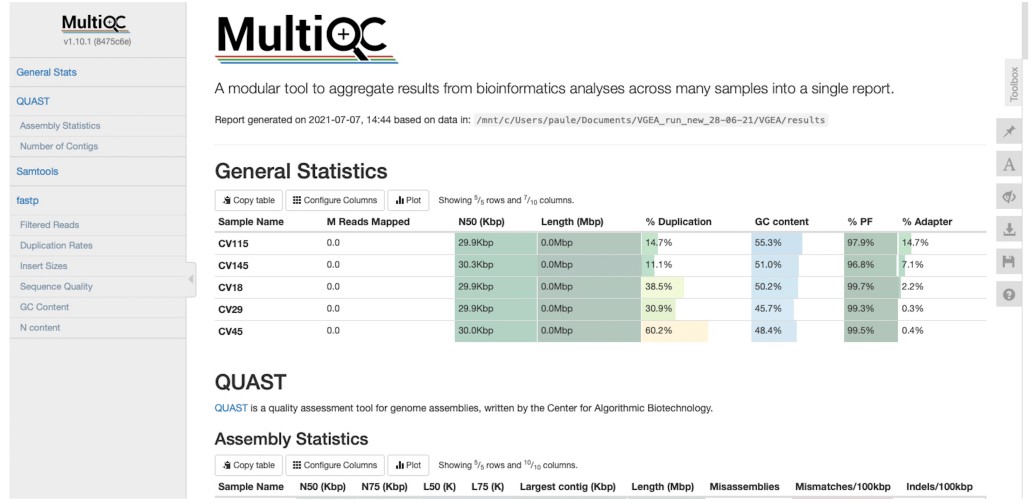

**Figure 3** MultiQC report of five SARS-CoV-2 datasets analyzed using VGEA.

**Table 1  Benchmarking values (time and CPU usage) for a SARS-CoV-2 dataset analyzed using VGEA.**

| VGEA rule name | Time (h:m:s) | Maximum RAM used (MB) |
|---|---|---|
| human_reference_index | 1:01:53 | 4688.56 |
| fastp | 0:00:14 | 581.91 |
| bwa_human | 0:08:52 | 5960.95 |
| samtools_extract | 0:02:40 | 16.21 |
| bamtofastq | 0:01:39 | 6.61 |
| [a]iva | 8:19:11 | 238.57 |
| shiver_init | 0:00:53 | 64.97 |
| shiver_align_contigs | 0:04:37 | 2509.64 |
| shiver_map_reads | 0:31:51 | 567.27 |
| shiver_tidy | 0:00:00 | 1.06 |
| quast | 0:00:33 | 72.51 |

**Notes.**

[a]IVA was run using one CPU core and two threads so if allowed more computational resources, the assembly time will be even shorter.

## Performance evaluation

VGEA makes use of Snakemake's benchmarking feature which allows the measurement of the CPU usage and wall clock time of each rule in the pipeline. This allows the user to know which step of the pipeline requires the least and highest amount of computational resources. Knowledge of this can help the user decide on the number of threads to dedicate to each rule as VGEA also makes use of Snakemake's multi-threading feature. Table 1 shows the benchmarking values for a sample SARS-CoV-2 dataset analyzed using VGEA.

We compared the contigs generated by VGEA's assembly step with contigs generated using two other standalone and commonly used assembly pipelines, SPAdes (*Bankevich et al., 2012*) and Velvet (*Zerbino & Birney, 2008*). We compared against these two pipelines because most commonly used assembly workflows like viral-ngs and VirAmp are built on them. We carried out this comparison by making use of five different SARS-CoV-2 test datasets (namely CV18, CV29, CV45, CV115 and CV145 datasets available on FigShare and NCBI). We compared the assemblies to the SARS-CoV-2 reference genome, and N50/NG50, mis-assembly, mismatches and indel scores were used to evaluate the performance of each assembly method as recommended by Assemblathon 2 (*Bradnam et al., 2013*) (Table 2). Basic statistics were calculated using QUAST. All results of our performance evaluation and comparison are provided as File S2. All analyses were run on a 64-bit personal computer with 16GB RAM using four threads. SPAdes version 3.15.2 and Velvet version 1.2.10 were used for the comparison purposes using the default parameters.

Evaluation statistics showed that contigs generated by VGEA had the highest NG50 score for four of the five datasets and the highest N50 scores across all five datasets. In all five datasets, VGEA's contigs had the highest genome fraction covering greater than 95% in four.

**Table 2 Performance comparison using different assembly pipelines.**

| Sample ID | # reads (x10^6) | Pipeline | # contigs | Largest contig (bp) | N50 | NG50 | Genome fraction (%) | Mis assemblies | Mismatches | Indels | Maximum RAM used (MB) |
|---|---|---|---|---|---|---|---|---|---|---|---|
| CV18 | 3.2 | VGEA | 42 | 29928 | 2294 | 29928 | 99.776 | 0 | 10 | 0 | 627 |
| | | SPAdes | 384 | 22141 | 1435 | 22141 | 99.652 | 1 | 18 | 1 | 2447 |
| | | Velvet | 68 | 1858 | 728 | 922 | 19.326 | 0 | 3 | 0 | 1544 |
| CV29 | 1.8 | VGEA | 31 | 7731 | 3065 | 7534 | 99.786 | 0 | 9 | 0 | 484 |
| | | SPAdes | 478 | 24904 | 1136 | 24904 | 99.632 | 0 | 7 | 0 | 2314 |
| | | Velvet | 66 | 2877 | 942 | 1380 | 1.729 | 0 | 0 | 0 | 807 |
| CV45 | 6.2 | VGEA | 30 | 16248 | 2603 | 16248 | 98.291 | 1 | 11 | 0 | 666 |
| | | SPAdes | 45 | 6779 | 1255 | 2447 | 94.957 | 0 | 35 | 12 | 2504 |
| | | Velvet | 535 | 5239 | 898 | 3030 | 14.256 | 0 | 0 | 0 | 1360 |
| CV115 | 2 | VGEA | 28 | 5225 | 2258 | 3060 | 96.957 | 0 | 12 | 0 | 177 |
| | | [a]SPAdes | 49 | 1942 | 1068 | 1828 | - | – | – | – | 1735 |
| | | Velvet | 41 | 2847 | 819 | 931 | 68.134 | 0 | 9 | 0 | 511 |
| CV145 | 4.4 | VGEA | 28 | 6807 | 2049 | 4214 | 73.093 | 0 | 14 | 0 | 635 |
| | | SPAdes | 188 | 3216 | 1190 | 2477 | 5.073 | 2 | 13 | 0 | 2547 |
| | | Velvet | 178 | 1798 | 682 | 1107 | 3.578 | 0 | 0 | 0 | 1459 |

**Notes.**

[a]QUAST gave no genome fraction value for this sample.

Comparison of maximum RAM used by VGEA, SPAdes and Velvet showed that VGEA used the least amount of RAM for the analyses of all five datasets used for comparison. SPAdes and Velvet however ran faster than VGEA for all analyses.

## DISCUSSION

VGEA is built on the snakemake workflow management system (*BKöster & Rahmann, 2012*), a workflow management system that allows the effortless deployment and execution of complex distributed computational workflows in any UNIX-based system, from local machines to high-performance computing clusters. It is a user-friendly, customizable and reproducible pipeline which can be deployed on a personal computer and which can run from start to finish with a single command.

VGEA was designed with ease-of-use in mind and so all its dependencies can be installed in a conda environment under the bioconda channel (*Grüning et al., 2018* making it particularly useful for scientists with little or no computational background and for scientists in LMICs who don't have much access to high-performance computing clusters or cloud-computing resources. VGEA capitalizes on Snakemake's multi-threading feature so that makes it possible for it to be deployed on laptops with greater computing performance or a computing server to improve its speed. The pipeline was tested with paired-end short-read sequencing data produced by the illumina platform (MiSeq, MiSeq FGx and HiSeq 2500).

The results generated by the major steps of the VGEA pipeline are summed up together into a MultiQC report which can be easily interpreted and understood by anyone with little or no knowledge of bioinformatics.

## CONCLUSION

VGEA was built primarily by biologists and in a manner that is easy to be employed by users without significant computational background. As new and innovative tools for viral genome analysis and assembly are increasingly being developed, these can easily be incorporated into the VGEA pipeline. We hope that other scientists can build upon and improve VGEA as a tool to extract more qualitative and quantitative information from viral genomes.

**Abbreviations**

| | |
|---|---|
| **VGEA** | Viral Genomes Easily Assembled |
| **NGS** | Next generation sequencing |
| **RNA** | Ribonucleic acid |
| **SARS** | Severe Acute Respiratory Syndrome |
| **MERS** | Middle East Respiratory Syndrome |
| **IVA** | Iterative Virus Assembler |
| **SHIVER** | Sequences from HIV Easily Reconstructed |
| **HPC** | High Performance Computing |

## ACKNOWLEDGEMENTS

We appreciate the continuous support of ACEGID staff and the management of Redeemer's University. We especially appreciate Dr. Finlay Maguire and Dr. Gerry Tonkin-Hill for helpful discussions and for making necessary changes to the pipeline. Also, thanks to Dr. Andreas Wilm and Christopher Tomkins-Tinch for helpful comments and suggestions.

### Funding

This work is made possible by support from Flu Lab and a cohort of generous donors through TED's Audacious Project, including the ELMA Foundation, MacKenzie Scott, the Skoll Foundation, and Open Philanthropy. This work was supported by grants from the National Institute of Allergy and Infectious Diseases (https://www.niaid.nih.gov), NIH-H3Africa (https://h3africa.org) (U01HG007480 and U54HG007480 to Christian T Happi), the World Bank grant (worldbank.org) (project ACE019 to Christian T Happi), and the Wellcome Trust grant (https://wellcome.ac.uk) (216619/Z/19/Z to Christian T. Happi and Jonathan L. Heeney), and the AAS grant SARSCov2-4-20-022 to Christian T. Happi. The funders had no role in study design, data collection and analysis, decision to publish, or preparation of the manuscript.

### Grant Disclosures

The following grant information was disclosed by the authors:
Flu Lab.
TED's Audacious Project.

ELMA Foundation.
MacKenzie Scott.
Skoll Foundation, and Open Philanthropy.
National Institute of Allergy and Infectious Diseases.
NIH-H3Africa: U01HG007480, U54HG007480.
World Bank grant: project ACE019.
Wellcome Trust grant: 216619/Z/19/Z.
AAS grant: SARSCov2-4-20-022.

## Competing Interests

Simon D.W. Frost is employed by Microsoft Research and is an Academic Editor for PeerJ. All other authors have declared that no competing interests exist.

## Author Contributions

- Paul E. Oluniyi conceived and designed the experiments, performed the experiments, analyzed the data, prepared figures and/or tables, authored or reviewed drafts of the paper, and approved the final draft.
- Fehintola Ajogbasile, Judith Oguzie, Jessica Uwanibe, Adeyemi Kayode, Anise Happi, Alphonsus Ugwu, Testimony Olumade, Olusola Ogunsanya and Philomena Ehiaghe Eromon conceived and designed the experiments, performed the experiments, authored or reviewed drafts of the paper, and approved the final draft.
- Onikepe Folarin, Simon D.W. Frost, Jonathan Heeney and Christian T. Happi conceived and designed the experiments, authored or reviewed drafts of the paper, and approved the final draft.

## Data Availability

VGEA is freely available on GitHub at: Available at https://github.com/pauloluniyi/VGEA under the GNU General Public License.

All primary test datasets used for the validation of the VGEA pipeline are available at figshare: Oluniyi, Paul; Ajogbasile, Fehintola; Oguzie, Judith; Uwanibe, Jessica; Kayode, Adeyemi; Happi, Anise; et al. (2020): VGEA: A snakemake pipeline for RNA virus genome assembly from next generation sequencing data. figshare. Dataset. Available at https://doi.org/10.6084/m9.figshare.13009997.

All SARS-CoV-2 and Lassa virus test datasets are available at NCBI SRA (BioProject: PRJNA666685 and PRJNA666664). All HIV-1 test datasets are available on NCBI SRA: ERR3953696, ERR3953853, ERR3953893, ERR3953891, ERR3953866, ERR3953846, ERR3953756, ERR3953877, ERR3953876, ERR3953750, ERR3953741, ERR3953697, ERR3953699, ERR3953706, ERR3953708, ERR3953710, ERR3953712, ERR3953716, ERR3953295, ERR3953693.

## Supplemental Information

Supplemental information for this article can be found online at http://dx.doi.org/10.7717/peerj.12129#supplemental-information.

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
