# Peer review of "VGEA: an RNA viral assembly toolkit"

_PeerJ, doi:10.7717/peerj.12129_

## Round 0.1 · original submission · Major Revisions

This work and manuscript is not ready for publication.

Many issues are raised concerning the evaluation by various reviewers.
Here, I specifically would like to point out the problems concerning the "craftsmanship" with the Snakemake implementation, so please consider the important comments of Reviewer 1:
- "The implementation is rather a pseudo Snakemake workflow as most of the steps are contained within quasi sub-pipelines implemented as standard shell scripts."
and
- "The authors should try to implement the Snakemake best practice project structure where the complete configuration can be performed in a single config file."
- Starting with BAM files is not an option for a full workflow.

So in fact, what you are presenting is not really a well-written configurable reproducible workflow. State-of-the art and best practices (please read about that in the Snakemake documentation) recommend that the workflow is configurable via a config file, delegates common tasks to subworkflows or existing wrappers, but otherwise keeps everything within the Snakemake system.

Therefore, the expected changes will indeed be "major".

Reviewer 1 ·

Basic reporting

The authors present a Snakemake pipeline for automated assembly tailored to RNA virus genomes.
In the manuscript they describe the individual steps of the analysis pipeline and present a comparison to the
'viral-ngs' pipeline from Broad Institute.

While the manuscript describes the most important steps of the analysis, there are also some important details missing:

- As one example, there is no description on how the final consensus sequence is derived from the mapping. While this is also important to interpret the comparison with the viral-ngs pipeline.
Are missing/uncovered positions masked with 'N' or simply filled with the best matching reference? The latter could lead to shorter assemblies only at the two end-points, as all gaps between contigs were previously filled with the best matching reference by shiver.
- With one central argument for using a workflow system like Snakemake is reproducibility, I also miss the summary of tool parameters that were used to generate the examples for the provided test sequences. Even more severe, it is not documented which sequences were used as references!
- Many tools listed in the dependencies (Fastp, KMC, Mummer, SMALT, BOWTIE, SPAdes) of the pipeline are not mentioned in the manuscript. So it is not clear how/why they are used.
- An overview figure with a flowchart of the complete workflow would help a lot!
- There is no real description how the user would configure and run the workflow, only how a Singularity container could be built.

Experimental design

- For the evaluation I believe that assembly length is not the only important parameter to be considered to evaluate the overall quality like, e.g. the error rate. Recurring to my previous point about the consensus sequence, it is important to be clear how exactly the genome length was defined, whether positions filled from the closest reference are also counted and if this is the same between for both approaches.
I further doubt that a statistical significance of genome length is some valid indicator without considering the qualitative significance of the missing genomic regions.
- The choice of bam files as mandatory entrance to the workflow is a suboptimal choice in my opinion. At least fastq (also gzipped) files should be an additional option. As the authors motivate the bam files by a frequently conducted preceding decontamination step, this should then rather be integrated to the workflow as an optional preprocessing step.

IMPLEMENTATION

- A real manual describing how to setup, configure and run the workflow is missing in the manuscript and also on the github page. There should be examples to explain the user how to configure the workflow and how to start it such that it will use the provided conda environment.

- While the option of using Singularity is documented, the alternative option of how to use the provided conda environment for the workflow is completely missing.
Also, there is no hint to the config file, which is rather hidden in the scripts folder with the confusing file ending ".sh".

- If I see it correctly, the paths to the references, primers and adapters need to be configured within the Snakemake file at the input section of the corresponding rule. The authors should try to implement the Snakemake best practice project structure where the complete configuration can be performed in a single config file.

- The implementation is rather a pseudo Snakemake workflow as most of the steps are contained within quasi sub-pipelines implemented as standard shell scripts.
Thus, the workflow does not make use of many Snakemake features, like, e.g. it does not use its capabilities to distribute the work over multiple cores.

-With the current state, I really doubt unexperienced users will be able to run the workflow.

Validity of the findings

no comment

·

Basic reporting

The authors present a well-written manuscript describing their development of a novel pipeline for the de novo assembly of viral genomes. While the text is well-structured and well written, the current state of the art was not sufficiently well described and relevant references (V-GAP, VirusTAP or VrAP just to name a few potentially competing tools) are missing.

In this vein, I strongly recommend describing the actual problems with viral assembly (such as the minoritary variant problem hinted at in several places) in and existing approaches in more detail. While the introduction does a very good job of describing why NGS and viral bioinformatics are valuable tools, in my opinion it leaves the reader wondering what the actual motivation behind the development of VGEA is (apart from a very broad "it helps assemble viral NGS data").

Minor points:

* Line 61: I would recommend using something like "open view" instead of "unbiased" - while I can see what the authors mean, I feel that this glosses over the (often problematic, also in viral NGS) bias introduced by library preparation in NGS experiments.

* Line 62: I recommend rephrasing this sentence. It is in my opinion unfortunate to directly compare NGS "for virus detection and assembly" with diagnostic methods, that are by design only meant for virus detection. Again, I entirely agree with the point the authors are making, however I find the phrasing unfortunate.

* Line 40: The conclusion could be expanded with an informative, summarizing sentence - I feel like there should be more to the conclusion than the (of course very important) github link.

Experimental design

While opinions on how original combining existing tools into a pipeline is, I think that this is highly valuable work given how many different tools exists. The resulting number of possibilities to combine them into pipelines that solve the same problem is staggering, and it has been well documented that the choice of tools for each step in an analysis can have a massive impact on the quality of the analysis results. I thus feel that the work presented in this manuscript is a valuable, original contribution to the field.

However, in my opinion several points in the description, motivation and evaluation leave significant room for improvement and clarification:

* Since a reference sequence covering gaps in the initial assembly using the closest reference sequence is created using Shiver (lines 144-146) and the final sequence is created by mapping to this sequence (lines 146-148), the potential bias introduced by this use of a reference sequence should be evaluated and the process of generating the consensus sequence from that mapping result should be described in more detail (e.g. what happens with uncovered regions, what happens in edge cases such as the reference being covered by only one or by conflicting reads). This can relatively easily be evaluated by creating simulated datasets, and I feel like this is an important step in ensuring that the final contigs are not biased towards the reference sequence in areas of low coverage or where assembly is hard due to sequence characteristics.

* It is very commendable that the authors have put thought and effort into making their pipeline easy to run and deploy using singularity. However, I would recommend either removing the claim of reproducibility from line 110 or adding an extra sentence describing how this is achieved. Specifically, in order to ensure reproducibility (which is an extremely desirable trait) two points are of major importance: (i) checking for the use of random seeds in the tools within the pipeline and allowing users to set those seeds from outside of the pipeline and (ii) dealing with parallelity: either ensuring that all used tools produce the same result independent of the order of the operations/input files (e.g. when performing mapping with multiple threads, the ordering of sam/bam files might not be reproducible unless special optionss are provided), or offering an option to the user that allows ensuring reproducible ordering (generally at the expense of computational efficiency)

* In the section "Mapping" (lines 138-150) it is not clear to me what the exact process and filtering steps are. Are reference sequences supplied by the user, or is an alignment of potential reference sequences required? Also, aligning contigs to reference sequences using MAFFT is potentially problematic: (i) In case of structural variations (e.g. recombinations, which in some viruses happen within the patient), MAFFT is unable to produce a clean alignment since it performs global alignment and (ii) as far as I know, default mafft is not especially good at dealing with end gaps (such as in contigs only covering part of the genome) - using mafft-linsi might be helpful, but the scripts in the repository use mafft.

Validity of the findings

he performance improvements over viral-ngs presented by the authors are impressive and the datasets used to perform validation of the results have been well documented. However, in my opinion several points require significant further work:

* Most importantly, using only reconstructed genome length as a quality metric does not do justice to the complexity of the task. An especially important metric is the number of errors. This ties into my comment concerning the risk of biases introduced by joining contigs using reference sequences. So, not only the length but also the error rate and types of the assemblies should be compared

* The choice of viral-ngs as the only tool VGEA was compared against is not sufficiently well motivated - there are several other tools that could be seen as competitors: V-GAP, VirusTAP or VrAP just to name a few.

* Again, tying into the bias comment: It is not clear whether the "true" reference sequence was excluded from the reference set provided to VGEA in the comparison. Given that VGEA uses the references to fill in gaps in the contigs, a realistic scenario would be knowledge of all reference sequences available up to the moment when the sequencing run was performed - providing VGEA with reference sequences published afterwards (in the worst case: those resulting from analysis of the NGS data used in the comparison) would provide VGEA with an unfair advantage

* How reads for the comparison were pre-processed should be made clear: Were the same trimmed and background-filtered reads used for viral-ngs as for VGEA?

I feel like only once these points have been adressed the statement "This significantly improves the quality of the genomes obtained from NGS data" in line 188 in the discussion can be made with confidence.

Additional comments

In this manuscript, the authors present a novel pipeline for reference-assisted de novo assembly of viral genomes. This is an important and task in bioinformatics that will profit from further improvements in the available tooling. However, while the manuscript presents a good structure and is well-written, in my opinion significant additional work is required to allow this promising work to achieve its full potential and to make it useful for the community.

Reviewer 3 ·

Basic reporting

The authors present a pipeline for viral genome reconstruction from NGS data. The pipeline is implemented in the form of a Snakemake workflow, which preprocesses, aligns and assembles the reads into a consensus genome. The manuscript is well written with perfect use of professional English. Several sequencing data sets were generated for assembly evaluation, these data sets are provided online. However, I do have some comments regarding the background information (introduction) provided:

1. Although viral diversity and quasispecies behavior is mentioned in the abstract, this is not discussed in the introduction. I believe this is an important aspect of viral genome analysis; the authors should expand on the difference between consensus assembly and haplotype assembly (strain-level reconstruction) and clarify the goals of their method in this respect.


2. The authors present a new pipeline for viral genome assembly, but existing assemblers and assembly pipelines are not discussed. There is a whole scala of methods for genome assembly of RNA viruses, as well as review articles evaluating these methods. See for example “Choice of assembly software has a critical impact on virome characterisation” by Sutton et al (2019).

Experimental design

The topic of the research is of high relevance, as viral genome assembly can be very challenging. Some comments regarding the experimental design:

1. The authors should state more explicitly the aims of their research, and explain how their research fills an existing gap. This extends my previous comment regarding the background information: there are many viral genome assembly tools and pipeline available, and it is not clear why these do not suffice. What is the gap that the authors try to fill with VGEA? I do not understand why the authors do not mention or evaluate a recent pipeline called V-Pipe, which I believe serves the same as VGEA.

2. In addition to a theoretical discussion of VGEA in comparison to available tools, presenting a new pipeline also requires benchmarking experiments. The authors compare only to a relatively old approach (viral-ngs) but do not present results for other pipelines. How well does VGEA perform in comparison to other methods?

3. The authors have generated many datasets to evaluate their method on, which is very valuable. However, to truly consider these data sets as benchmarks, there should also be a "ground truth". For example, by generating high quality reads with Sanger sequencing, or PacBio HiFi sequencing. Currently, the authors only evaluate contig length, but this is too limited. Also the assembly accuracy (target reconstructed, error rate, misassemblies) should be considered. Given a ground truth sequence, these statistics can be quickly generated with an assembly evaluation tool like QUAST. It would also be interesting to see how the assemblies compare to the available reference genome.

4. I got very confused in the methods section. The pipeline starts with the splitting of BAM files, but it is not clear how these BAMs are generated. The text suggests that these BAMs are alignments to the human reference genome? Does that mean that VGEA uses all unaligned reads to perform assembly? Later, it is mentioned that reference sequences are used to remove contaminants; how exactly does this work, and which references did the authors use? Finally, the pipeline ends with Shiver creating a reference sequence and reads are mapped to this sequence. But then what is the purpose of these alignments? All of this is not clear from the text, it would really help me if additional details were added, along with a figure that illustrates the entire workflow.

5. The assembly step is performed with IVA. The authors state that "it has been demonstrated to outperform all other virus de novo assemblers". However, IVA was published in 2015 and many other assemblers have been developed since. At this point, there are probably better choices available, and it is important to see how the choice of assembler affects the results of the VGEA pipeline.

Validity of the findings

The software and data sets are available online, but I was unable to find the command lines / config files to enable full reproducibility.

I believe that a more extensive experimental evaluation (see previous comments) would also allow for deeper, more confident conclusions.

---

## Round 0.2 · Major Revisions

As you can see, all reviewers acknowledge that the workflow and the manuscript have considerably improved, but also each of them lists a number of points that should be further addressed. Taken altogether, this requires another rather major revision, even if some of the comments just address minor usability issues.

Reviewer 1 ·

Basic reporting

Minor:
- Tool / software name typos (capitalization) should be fixed e.g. snakemake (Snakemake), quast (QUAST).
- Some tools are cited multiple times (e.g. shiver three times).

Experimental design

While the authors transformed their workflow into a "real" Snakemake workflow, there are still some features missing and things to be improved:

=== IMPLEMENTATION ===

- While the authors state that "VGEA capitalizes on Snakemake’s multi-threading feature", they are not using the "threads" feature of Snakemake in any of their rules.

- The authors claim that "VGEA allows full customization of the pipeline, so
145 users can modify the parameters used in running their samples". However, the only step to be controlled by the user directly is shiver via its separate config file. For all other rules parameterization via the "params" section of Snakemake together with an augmented config.yaml is NOT possible.

- While the entry point are now the fastq files and the authors mention Trimmomatic and fastp, I think these tools should also be added to the workflow. Shiver (depending on its config file) will apply Trimmomatic but iva runs on untrimmed/filtered reads. While iva could be configured to run Trimmomatic too, the most efficient way would be an explicit trimming step at the very beginning of the workflow together with a quality control. Finally, an aggregation of these results together with the results from QUAST in a single report (multiQC) would be another plus.

- The authors should use the proper mechanism to use Conda with Snakemake, where Snakemake takes care of the environment creation, activation/deactivation instead of letting the user create a global Conda environment first - as proposed on the GitHub readme page.

- Input: Currently the workflow will use all fastq files fulfilling the naming pattern something_R1.fastq something_R2.fastq located in the working directory. The authors could enhance the usability and flexibility and drop this strict naming constraint if they would use the typical samples/units.tsv scheme to be found in all best practices Snakemake workflows.

=== DOCUMENTATION GIT / WEBSITE ===

- The authors should specify which output files will be generated, where they will be found and what information they contain. Probably with a small example run presented on the readme page.

=== MANUSCRIPT ===

- I think the statement "VGEA generates a number of files among which is a CSV file that contains minority-variant information (base frequencies at each position), consensus genomes and several quality metrics to assess the genomes assembled." is somehow misleading. To me this suggests that these files are produced by VGEA in some new, additional downstream analysis step. If I am correct, these files are produced by shiver directly - this should be clarified.

Validity of the findings

no comment

Additional comments

no comment

·

Basic reporting

The authors present a manuscript that has been significantly reworked. It is clear that the comments have been taken seriously and the authors have made a significant effort to address the points that have been raised.

However, there remain a few (minor) points that should to be adressed:

* Lines 115-119: "Most"/"A lot" are redundant with "usually", I recommend dropping one of these from each of the two sentences here.
* Lines 121-124: The authors first describe how daunting a task it is to choose the correct tool from the massive list of available tools, and then go on to describe how they solve "the challenges listed above" by implementing yet another viral assembly pipeline. This is a somewhat unfortunate combination - I recommend just removing the sentence "It can also be overwhelming to pick a suitable software from the huge amount of bioinformatics tools available to carry out various tasks".
* Lines 115-124: There exist both pipelines that are specifically designed to run on personal hardware (e.g. V-Pipe specifically states how it allows choosing tools for each of the steps depending on available computational resources) and to make dependency management easier (e.g. this is a quote from the viral-ngs documentation: "'./easy-deploy-viral-ngs.sh setup' Installs a fresh copy of viral-ngs, installs all dependencies [...]"). It would be helpful if something like a table listing the cited tools and the requirements mentioned in the text was included, with checkmarks/crosses showing which of the tools satisfies which of the requirements. This would allow a quick and clear understanding of the gap that VGEA fills, rather than the general arguments saying that "most" tools "usually" don't fulfill the described requirements.

Experimental design

It is very commendable that the authors have updated their pipeline and the README to better address issues with reproducibility and runnability. However, significant problems remain. Specifically:

In README.md:
* All dependencies could be in one "conda install -c bioconda" line, instead of splitting them across multiple lines
* A description of how to exactly run on a test dataset (e.g. "snakemake --cores n -d data/LASV_L") and what the expected outputs are would be great
* There is no description on how to run the singularity image, a sample command would be very useful
* Having a singularity image as a downloadable artifact for those who are not admins but have permission to run singularity images would be helpful


In the Snakefile itself: After following the instructions in README.md, running the pipeline produces the error:

SyntaxError in line 8 of /home/wojtek/temp/VGEA/Snakefile:
EOF in multi-line statement (Snakefile, line 8)

One reason is that closing quotation marks are missing in the rule in lines 43, 52 and 130, but there are more syntax errors that I have not managed to find (the error persists even after closing these parantheses). These errors need to be fixed before publication. Additionally, in order to generally avoid such problems, I highly recommend creating a minimal test dataset and introducing a github action to make sure that commits do not fundamentally break the pipeline's functionality.


Also, the inclusion of a thorough description of the shiver method has indeed been helpful. However, this raises the issue of the required input files - given the criticality of the quality of the reference alignment for shiver, I feel that this (especially the sentence "producing such an input by automatically aligning a large number of diverse sequences without checking the results would be a bad idea" from the supplementary material) should be explicitly mentioned in the README. This may seem obvious to bioinformaticians but is especially important given the stated goal from the conclusion (lines 210-211): "VGEA was built primarily by biologists and in a manner that is easy to be employed by users without significant computational background".

Validity of the findings

Unfortunately, the authors seem to have taken my comments concerning the choice of tools (in my opinion too few) and metrics (in my opinion non-optimal) used to compare VGEA against other pipelines as motivation to entirely remove any comparison of VGEA's performance to that of other similar tools. This is highly unfortunate, as this leaves "it's easy to install and run" as the only argument for choosing VGEA - and this is a poor motivation for the choise of a tool to do research with. I thus believe that the comparison needs to be reintroduced and extended (using more tools and better metrics) rather than being removed entirely.

In the current state, the validity of the findings can thus not really be assessed.

Additional comments

The authors have significantly reworked the manuscript and improved it in many places. However, I believe that especially two major points must be adressed before publication:

* The pipeline needs to actually run
* Some form of performance comparison against existing tools needs to be shown in order to allow users to make an evidence-based (rahter than just convenience-based) decition

Reviewer 3 ·

Basic reporting

I would like to thank the authors for their detailed responses to my questions. Several things have been clarified, the background has been information has been extended and the goal of the manuscript has become much clearer: to design a pipeline for viral genome analysis that can run on a personal computer.

Experimental design

However, I still have one major issue with this manuscript: where are the results? One of the issues with the previous version of this manuscript was that the assembly results were not evaluated adequately, but the current version has no assembly results at all...

If the research question is "how to design a pipeline to do viral genome analysis on a PC?", the manuscript should *at least* present results on runtime and memory usage for this pipeline, as well as other existing ones. Do other pipelines, e.g. V-Pipe and viral-ngs, indeed require too much resources? And how do other pipelines compare in terms of assembly quality?

The figures and tables shown in the manuscript are literally just the QUAST output on one random sample, with no interpretation. I don't see the added value of this and I would recommend the authors to make their own figures and tables, showing results on resource usage and assembly accuracy.

Validity of the findings

The authors conclude "VGEA was built primarily by biologists and in a manner that is easy to be employed by users without significant computational background". Although the authors have improved the Snakemake workflow substantially, I still don't believe this is true in its current state. The manual provided describes extensively how to install conda and to create an environment with all dependencies, but the only instructions regarding the pipeline itself are:

"Run the snakemake pipeline: snakemake --cores n -d /workingdir/
The location of the human reference genome, the reference, primer and adapter fasta files for shiver and the reference and gene features files for quast can be controlled by adjusting the config.yaml in the VGEA working directory."

If users have to review the manual of the individual tools (bwa, iva, shiver) to set appropriate parameters, I don't see how this is any easier than just running the 3 tools separately. If the authors really want this to be an easy to use pipeline, it would be very helpful if there is more documentation on how to choose parameter and possibly also including a small test case.

Additional comments

line 85: “overlapping region in the genomes”. I think you mean overlapping sequencing reads?
line 120: what is AVL?
line 156-160: I think it is important to indicate here that the superiority of shiver was shown by the authors of shiver, not an independent review.

---

## Round 0.3 · Minor Revisions

As you can see, all reviewers appreciate the continued effort that went into the manuscript and the tool. Several minor issues remain, and I am confident that they can be satisfactorily resolved; so we can ultimately proceed with the publication.

Reviewer 1 ·

Basic reporting

Thanks to the authors for their responses to my previous concerns and the implementation of the missing features (conda, sample sheet, threads).

However, I still have some issues with the manuscript and the workflow documentation.

Experimental design

In the benchmark section, I feel there is some information missing and also the comparison should be extended:

- As computational efficiency is one major argument, I would like to see the resource footprint for one of the other pipelines mentioned on the same samples to substantiate the central claim that VGEA empowers scientists with very limited hardware.

- For the presented benchmark I miss some crucial information:
- Hardware specification!
- Which reference alignment has been used for shiver?
- How exactly were SPAdes and velvet run? Versions? Parameters? Were they run on raw reads or on the filtered and trimmed reads also used as input for IVA and shiver?
- Why does the shiver output (which, as I understand is the main result of the pipeline) contain multiple contigs? As I understood shiver would always return a contiguous full length assembly.
- What is the '-' in Table 2 for SPAdes on sample CV115?
- The fact that NG50 is always greater than N50 should be addressed. Does that mean, that all assemblies contain a lot of junk/contamination or duplicated segments?
- Which reference was selected for QUAST and why?

Some minor issues in the Readme.md
- The listing of the results file tree does not contain shiver for test1 (which should rather be renamed sample1 for clarity), hence it is not described.
- The given command for a test run gives a wrong path for the test_config.yaml. The folder is .tests (not .test as given in the Readme)

Some problems when running the test example:
- I received an error for IVA due to some conda/mamba issues (ModuleNotFoundError: No module named 'pkg_resources'). For me it could be fixed by adding "setuptools==52.0.0" and "kmc=3.1.1rc1=h76f5088_0" to iva.yaml.

- The same problem also leads to an error for shiver (Error running 'fastaq version'. Are you sure that fastaq is installed, and that you chose the right value for the config file variable 'fastaq'?). Could be fixed again by adding "setuptools==52.0.0" to shiver.yaml

Validity of the findings

no comment

Additional comments

It is good to see how the quality of the implementation improved in the two iterations. If the remaining points in the manuscript will be addressed it should be ready for publication.

·

Basic reporting

The authors present a manuscript that has again been significantly reworked. It is clear that the authors have gone to great lengths to seriously address all issues that were raised in the previous reviews.

The only remaining point I see is the availability of the data.

Firstly, there is a typo in the data repository: The link in line 131 (https://doi.org/10.6084/m9.figshare.13009997) is correct, however the one mentioned in the data availability statement (https://doi.org/10.6084/m9.figshare.1300999) is wrong (leads to a "DOI Not Found" error page). I suspect that this is supposed to be the same link and that is is just a typo.

Secondly, in lines 192-193, it is mentioned that "We carried out this comparison by making use of five different SARS-CoV-2 test datasets." However, it is not clear which of the 63 datasets available under the figshare link have been used for the evaluation. This should be made clear.

Experimental design

Again, the previous concerns have been very well addressed. I am especially happy to see that automated testing using git actions has been implemented to ensure that the software runs correctly. A quick test confirms that the previously present bugs have been resolved.

One minor point: In the two commands at the end of README.md:

snakemake --use-conda --use-singularity --configfile .test/integration/test_config.yaml -j 1

and

snakemake --use-conda --configfile .test/integration/test_config.yaml -j 1

the test folder has a typo - it should be ".tests" and not ".test".

Validity of the findings

While I am very pleased to see that benchmarking has been re-introduced, I am still not entirely satisfied with how it was performed.

Firstly, resource use and runtime were evaluated for the rules used in VGEA, but there is no comparison of resources use and runtime of VGEA as a whole as compared to the other initially mentioned pipelines. In the introduction, V-Pipe and viral-ngs are specifically mentioned as tools that require too much computational power and/or are too hard to install and thus motivate the development of VGEA. It would be advisable to demonstrate at this point that VGEA actually needs fewer resources or is quicker that these tools (since both provide installation instructions that are similar in simplicity to those of VGEA).

Also, the comparison of assembly results against velvet and spades seems oddly chosen, since velvet and spades are assemblers that are used in pipelines, not end-to-end pipelines in their own right. As such, it would seem more intuitive to compare the results of VGEA against the results of V-Pipe or viral-ngs - unless the performance comparison shows that they indeed need much more resources and VGEA is the only viable option on the used hardware, in which case this should be explicitly mentioned as the rationale for comparing VGEA with these assemblers.

In either case, just contig length/number metrics are in my opinion insufficient. In my opinion, at the very least the misassembly, mismatch and indel metrics from QUAST should be included in this comparison. While it is true that Bradnam et al. recommend using genome coverage as a comparison metric (as mentioned in the manuscript as motivation for using this as a basis for the comparison), it is only one dimension of the COMPASS metrics used and recommended in the cited paper. They even have a section titled "Size isn't everything" in their Discussion and specifically say that " we find that N50 remains highly correlated with our overall rankings. However, it may be misleading to rely solely on this metric when assessing an assembly's quality".

As such, I still find it hard to judge the validity of the findings (or specifically, whether VGEA actually delivers what is described in the introduction as the motivation for its development) based on the data presented here.

Reviewer 3 ·

Basic reporting

The authors have made an impressive effort and addressed my concerns satisfactorily. My only remaining comment is that, in the performance analysis, the assembly statistics evaluated are all related to assembly contiguity and completeness, but not accuracy. For the datasets evaluated no ground truth is known, which explains why accuracy is not evaluated. However, improved contiguity often comes at the expense of reduced accuracy. I therefore believe that it is important to add one sentence to the analysis of the assembly results, pointing out that accuracy was not evaluated, such that readers are aware of this potential limitation.

Experimental design

No comment.

Validity of the findings

No comment.

---

## Round 0.4 · accepted · Accept

thank you for the continued improvements of your manuscript. While one could continue to argue about some details of implementation (and some reviewers might), we think that the work is technically sound and useful to the community and in a good state to be published as it is.